# Effects of Age and Playing Position on Field-Based Physical Fitness Measures in Adolescent Female Netball Players

**Daniel A. Hackett [1],\*** , **Derek L. Tran [1,2,3]** , **Kimberley L. Way [4,5]** and **Ross H. Sanders [1]**

1 Discipline of Exercise and Sports Science, Sydney School of Health Sciences, Faculty of Medicine and Health, The University of Sydney, Camperdown 2006, Australia; derek.tran@sydney.edu.au (D.L.T.); ross.sanders@sydney.edu.au (R.H.S.)

2 Central Clinical School, The University of Sydney School of Medicine, Camperdown 2006, Australia

3 Department of Cardiology, Royal Prince Alfred Hospital, Camperdown 2006, Australia

4 Institute for Physical Activity and Nutrition, School of Exercise and Nutrition Sciences, Deakin University, Geelong 3125, Australia; kim.way@deakin.edu.au

5 Division of Cardiac Prevention and Rehabilitation, University of Ottawa Heart Institute, Ottawa, ON K1Y4W7, Canada

\* Correspondence: daniel.hackett@sydney.edu.au

**Abstract:** This cross-sectional study investigated the impact of age and playing position, controlling for maturity, on physical fitness indicators in 303 adolescent female netball players aged 12.0 to 15.9 years. Assessments included estimated maximal oxygen uptake ($VO_2max$) via the 20 m shuttle run test, 10 m and 20 m sprints, change of direction speed (CODS) using the 505 test, and muscle power via the medicine ball chest throw (MBCT) and countermovement vertical jump (CMJ). Participants were grouped by age (12 to 15 years) and playing position (non-circle and circle players), with age at peak height velocity as a covariate for maturity. Results revealed that, at 15 years, CMJ height was greater than at 12 years and 13 years ($p < 0.05$, partial $\eta^2 = 0.048$). MBCT distance increased across age groups ($p < 0.01$, partial $\eta^2 = 0.323$). Age had no impact on sprints, $VO_2max$, or CODS. Non-circle players outperformed circle players in the 10 m sprint ($p = 0.042$, partial $\eta^2 = 0.016$) and 20 m sprints ($p = 0.010$, partial $\eta^2 = 0.025$) and displayed higher $VO_2max$ ($p < 0.001$, partial $\eta^2 = 0.036$). Circle players were taller ($p = 0.046$, partial $\eta^2 = 0.014$) and heavier ($p < 0.001$, partial $\eta^2 = 0.040$) than non-circle players. Playing positions showed no differences in CMJ and MBCT. In adolescent female netball players, only muscle power is influenced by age, while non-circle players exhibit superior aerobic fitness and speed compared to circle players. Coaches may be able to utilize the distinct age and playing position traits of adolescent netballers to inform player selection and design targeted training programs.

**Keywords:** sprint; vertical jump height; muscle power; aerobic capacity; team sport; performance testing; maturation

## 1. Introduction

Netball is a popular team sport among females that draws participation from approximately 20 million individuals across 80 nations [1]. The sport involves intermittent activity characterized by alternating phases of vigorous exertion (such as sprinting, jumping, and directional changes) and short intervals of rest or reduced-intensity actions like walking, standing, and passing [2–5]. A standard netball match is played over four 15 min quarters on a compact court with dimensions of 30.5 m in length and 15.25 m in width (an area of 465 square meters). The demands of netball are multifaceted, encompassing skill, strategy, speed, agility, strength, and power [6]. Additionally, the energy needed for performance is derived from both aerobic and anaerobic sources. However, playing position has been shown to influence the physiological demands and activities during a game of netball [4,6,7]. There are seven playing positions in netball, which consist of midcourt roles (center [C],

wing attack [WA], and wing defense [WD]), shooting roles (goal shoot [GS] and goal attack [GA]), as well as defensive roles (goal keep [GK] and goal defense [GD]).

Physical fitness testing is pivotal for youth athlete training and aids in talent identification [8]. For netball players, testing can assist with individualizing training programs to target specific physical qualities essential for successful netball performance [6]. Analyzing 85,347 fitness test results from Australians aged 9 to 17 years, performance exhibited a positive trend linked to age, peaking around 15 years [9]. However, other studies have indicated that certain fitness aspects in female adolescents seem to necessitate at least 24 months for notable changes [10,11]. Thomas et al. [12] explored the physical fitness profiles of English regional academy netball players across age categories (under 15 years, under 17 years, and under 19 years). Age notably impacted jump height, sprint time, change of direction speed (CODS), and cardiorespiratory fitness. However, limited research has explored the impact of age on physical fitness in younger adolescent netball players, especially within 12-month intervals. Comprehending fitness trends among narrower age groups can crucially enhance youth netball player development, refine training tactics, and mitigate injury possibilities.

There is evidence of distinct physiological demands imposed on netball players across positions [4,7,13]. Center court players exhibit more frequent multidirectional movements and cover longer distances than GK and GS [7]. The longest shuffling durations have been observed in GD and WD [14], implying position-specific attribute preferences. Investigating youth netball players, McKenzie et al. [11] categorized them into circle (GK, GS, GA, GD) and noncircle (C, WA, WD) groups, since netball court constraints are a major factor impacting physical requirements (3). Non-circle players demonstrated superior sprint speed, CODS, and counter-movement vertical jump (CMJ) height, resulting in superior performance across diverse fitness tests [3]. Moreover, circle players, compared to non-circle players, were taller. The authors theorized that at 13 years old, more physically capable players might opt for non-circle positions, while for taller players, they might confine them to positions close to the goal. Notably, the study by McKenzie et al. [11] focused exclusively on 13-year-old females, with limited coverage of aerobic fitness and upper body muscle performance assessments. Expanding upon these findings to a wider age range and a greater variety of fitness measures could provide deeper insights concerning the physical fitness profiles of adolescent netball players.

This study aimed to evaluate how age and playing position influence aerobic fitness, sprint performance, CODS, and muscle power in adolescent female netball players, considering maturity. We hypothesized that older groups would exhibit superior fitness, with no differences between consecutive age groups (i.e., 12 months apart). Additionally, we predicted that non-circle-position players would outperform circle-position players in all the physical fitness measures. The study outcomes hold significance for coaches and sports scientists in player selection, training, coaching, and overall player development.

## 2. Materials and Methods

### 2.1. Study Design

To address the objectives of the study, a cross-sectional research design was employed. The participants were recruited from sports high schools in Sydney, New South Wales, Australia, and the data collection phase spanned from February 2018 to December 2020. Due to logistical constraints related to school schedules and facilities, the testing was spread over 1 to 3 days. The testing of participants took place either within specific classes or as part of team-based training sessions. The physical fitness assessment battery encompassed several performance measures: the 505 test was utilized to gauge change of direction speed (CODS), while the 10 m and 20 m sprints were conducted to evaluate speed. Aerobic fitness was assessed through the 20 m shuttle run test (20MSRT). Additionally, lower body muscular strength and power were measured using the countermovement vertical jump (CMJ), while upper body muscular strength and power were evaluated through the medicine ball chest throw (MBCT). To mitigate the potential impact of fatigue on test results,

strategic sequencing was employed. In instances where multiple tests were administered on a single day, the 20MSRT was consistently conducted last. This sequencing aimed to minimize any negative influence of fatigue on the performance outcomes of the other tests. Participants completed at least one practice attempt of the tests to become familiarized with them prior to commencing the actual attempts. Individual differences in maturity levels during data analysis were accounted for through calculating age at peak height velocity (APHV) [15] and subsequently utilizing it as a covariate to control for potential confounding effects related to maturation.

### 2.2. Participants

The study involved a cohort of 303 female netball players, ranging in age from 12.0 to 15.9 years. These individuals were recruited to partake in the research. Participants were requested to indicate their primary playing position in netball. In cases where participants specified multiple positions, the first position they mentioned was treated as their definitive position for the purpose of data analysis. Before their involvement in the study, explicit written consent was obtained from the parents (or legal guardians) of all participants. To be eligible to enroll in the study, participants needed to be healthy, which was deemed to be the absence of any musculoskeletal conditions or diseases. This was the primary responsibility of the educators from the high schools; however, researchers also conducted verbal pre-screening prior to tests to ensure that any risks to the health and safety of participants were minimized. The study received approval from the University of Sydney Human Research Ethics Committee (2015/878).

### 2.3. Anthropometry

Anthropometric measurements were conducted using standardized procedures. Stature was determined through the 'stretch stature method' using a stadiometer. Participants stood against the stadiometer, aligning their heels, buttocks, and back while ensuring that the upper border of the ear opening and the lower border of the eye socket formed a horizontal line. After stretching upward and holding a full breath, the headboard of the stadiometer was adjusted until it firmly touched the vertex of the head. Stature was measured in centimeters, with at least two measurements taken for each subject. Similarly, sitting height measurements were obtained with the participant seated on a wooden box in front of the stadiometer. Body mass was recorded using clinical scales graduated to 0.1 kg, with at least two trials for each participant. Height, body mass, and sitting height were entered into an Excel spreadsheet that was created using the Mirwald et al. [15] equation to calculate age at peak height velocity (APHV).

### 2.4. 505 Test

The CODS was evaluated using the 505-test, employing two timing gates (Fusion Sport, Coopers Plains, Australia) positioned 5 m away from a designated turning point. Participants initiated the test from a 2-point standing start position, 10 m from the timing gates (and subsequently 15 m from the turning point). They were instructed to accelerate rapidly through the 10 m timing gates, pivot at the 15 m line, and promptly return through the same timing gates. A standardized warm-up was conducted prior to the trials, encompassing a light jog (4 laps of basketball court), high knees, butt kicks, and lunges to half court, followed by a jog back. Following the warm-up, participants underwent three practice trials before proceeding to the actual test. Subsequently, they completed 2–3 timed trials (depending on time constraints), with a 3 min rest interval between each attempt. The validity of each trial required the turning foot to be placed on or over the turning line, verified by an assessor stationed at the 15 m mark. Participants were not given specific foot-turning instructions (i.e., left or right). In cases of trial invalidity, participants were permitted a 3 min rest before reattempting. The recorded time from the fastest trial, rounded to the nearest 0.01 s, was used as the 505-test score. Notably, the 505 test has exhibited robust test–retest reliability among high school athletes, characterized by an

intraclass correlation coefficient (ICC) of 0.88 (95% confidence interval: 0.81 to 0.93) and a coefficient of variation (CV%) of 2.4 [16].

### 2.5. Twenty-Meter Sprint Test

The assessment of maximum running speed involved a 20 m sprint using the 'Smart Speed' timing gate system developed by Fusion Sport, located in Coopers Plains, Australia. This system utilizes a single beam with error correction processing. The height of the photocells was adjusted to ensure that the beam would break at the participants' torso, in accordance with the manufacturer's instructions. The timing gates were positioned strategically at 10 m and 20 m from a predetermined starting point. Before performing the maximal sprints, participants underwent the standardized warm-up as previously described, which included submaximal sprints. Following this, participants completed 2–3 maximal sprints within time limitations, with a 3 min rest between each attempt. Each sprint started from a 2-point standing position, with the front foot placed 30 cm behind the start line and the initial timing gate. Participants were instructed to sprint as fast as possible over the 20 m distance from this standing start position. Timing was recorded to the nearest 0.01 s, and the fastest time achieved from the trials was recorded as the sprint score. Notably, the 10 m and 20 m sprint tests have exhibited strong test–retest reliability among junior athletes, with a CV% ranging from 1.82 to 3.05 [17].

### 2.6. Twenty-Meter Shuttle Run Test (20MSRT)

For sessions where only the 20MSRT was conducted, participants followed the previously outlined standardized warm-up routine. During the 20MSRT, participants ran back and forth between two marked lines placed 20 m apart on the ground. The running pace was synchronized with pre-recorded beeps played from an audio device connected to speakers. The interval between beeps gradually shortened, increasing running speed. The initial speed was 8.5 km/h, rising by 0.5 km/h each minute. A completed shuttle consisted of a successful run back and forth between the lines. Participants could miss one shuttle but needed to catch up on the next. If two consecutive shuttles were missed or participants chose to stop, the test ended. Scores were based on the total number of completed shuttles. Notably, the 20MSRT exhibited strong test–retest reliability among adolescents, with an ICC of 0.89 [16]. Maximal oxygen uptake (VO$_2$max), measured in ml·kg$^{-1}$·min$^{-1}$, was estimated using the equation validated by Léger et al. [18].

### 2.7. Countermovement Vertical Jump (CMJ)

Participants were instructed to stand with feet shoulder-width apart and hands on hips. A countermovement involved flexing the knees and hips before jumping upward. In flight, knee extension was maintained, but upon landing, participants were told to allow knee flexion to absorb impact. Jumps were performed on a contact mat (Smart Jump; Fusion Sport, Coopers Plains, Australia). Flight time (t) was measured to estimate the body's center of gravity height (h = gt$^2$/8, where g = 9.81 m·s$^{-2}$). A practice attempt was conducted before 1–3 trials, with at least 1 min rest between each. Verbal encouragement was given for maximal effort. The greatest CMJ height was used for data analysis. The CMJ has demonstrated robust test–retest reliability among adolescents, with an ICC of 0.96 [19].

### 2.8. Medicine Ball Chest Throw (MBCT)

An 8 m measuring tape was placed on the floor with its zero mark at the wall-floor junction. Prior to actual throws, participants extended their arms and dropped a 3 kg medicine ball to record the reach distance. Participants sat on the floor by the tape, legs extended, and back against the wall. They forcefully threw the ball away from their chests at a 30–45-degree angle, keeping their heads, shoulders, and lower backs in contact with the wall. Starting with the ball against their chest, they aimed to finish with their hands at nose height. After a practice attempt, 2–3 trials were performed with a 30 s recovery, with each throw's distance recorded to the nearest 0.1 m. If the form deviated or the trajectory

was incorrect, additional trials were performed. The result was throwing distance minus reach distance, with the greatest distance used. Researchers observed from the side, close to participants, and recorded accurately. Verbal encouragement was consistently given during each trial for motivation. The greatest MBCT distance was used for data analysis. In healthy undergraduate students, the MBCT has exhibited strong test–retest reliability, with an ICC of 0.97–0.99 [20].

### 2.9. Statistical Analyses

Data were presented as means $\pm$ SD and analyzed with SPSS version 28.0 for Windows (IBM Corp., Armonk, NY, USA). Participants were divided into five age groups: 12 years (12.0–12.9 years), 13 years (13.0–13.9 years), 14 years (14.0–14.9 years), and 15 years (15.0–15.9 years). Participants were categorized into two positional groups: non-circle and circle players [3]. Non-circle players encompass midcourt positions such as center, wing attack, and wing defense. Circle players consist of shooting positions, namely goal shoot and goal attack, as well as defense positions, including goal keep and goal defense. A two-way analysis of covariance (ANCOVA) was employed to assess the combined effects of age and playing position on each fitness parameter (CODS, 10 m and 20 m sprints, $VO_2max$, CMJ, and MBCT), adjusted for maturity (APHV) [15]. Post hoc tests, corrected using Bonferroni, were used for significant interaction effects to ascertain differences among age groups and playing positions. The effect size was quantified using the partial eta-squared ($\eta^2$) value [21]. Effect sizes were calculated via partial eta-squared ($\eta^2$), categorized as small ($\eta^2 = 0.01$ to <0.06), medium ($\eta^2 = 0.06$ to <0.14), and large ($\eta^2 \geq 0.14$). Statistical significance was set at $p < 0.05$ for all analyses [22].

## 3. Results

### 3.1. Effects of Age on Participant Characteristics and Performance

Significant age effects were observed for height ($p < 0.001$, partial $\eta^2 = 0.693$), with post hoc analyses indicating progressively greater height across successive age groups ($p < 0.001$) (Table 1). Likewise, body mass demonstrated a significant age effect ($p < 0.001$, partial $\eta^2 = 0.476$), with post hoc analyses reflecting a similar pattern of increasing body mass across consecutive age groups ($p < 0.001$) (Table 1). In terms of CMJ, an age effect was identified ($p = 0.007$, partial $\eta^2 = 0.048$), and post hoc analyses demonstrated greater CMJ height at 15 years in contrast to 12 years ($p = 0.045$) and 13 years ($p = 0.015$) (Figure 1A). Similarly, for MBCT, there was a significant age effect ($p < 0.001$, partial $\eta^2 = 0.323$), with post hoc analyses illustrating increasing MBCT distance across consecutive age groups ($p < 0.01$) (Figure 1B). There were no significant age effects for $VO^2max$, the sprints, or CODS.

**Table 1.** Participant characteristics and performances across age groups controlled for maturity based on age at peak height velocity.

| | Age Group | | | | Age Effects |
| | 12 Years | 13 Years | 14 Years | 15 Years | |
|---|---|---|---|---|---|
| n | 88 | 87 | 79 | 49 | |
| APHV (y) | 11.61 ± 0.47 | 11.79 ± 0.45 | 12.08 ± 0.39 | 12.42 ± 0.37 | |
| Height (cm) | 159.99 ± 3.33 | 165.34 ± 3.18 | 170.94 ± 3.19 | 175.49 ± 3.50 | $p < 0.001$; partial $\eta^2 = 0.693$ * |
| Body mass (kg) | 52.34 ± 10.69 | 61.85 ± 10.19 | 73.86 ± 10.24 | 84.19 ± 11.24 | $p < 0.001$; partial $\eta^2 = 0.476$ * |
| n | 76 | 77 | 67 | 42 | |
| CODS (s) | 2.81 ± 0.22 | 2.80 ± 0.21 | 2.77 ± 0.20 | 2.77 ± 0.24 | $p = 0.706$; partial $\eta^2 = 0.006$ |
| n | 79 | 81 | 70 | 44 | |
| 10 m sprint (s) | 2.05 ± 0.12 | 2.05 ± 0.12 | 2.03 ± 0.12 | 2.01 ± 0.13 | $p = 0.409$; partial $\eta^2 = 0.011$ |
| n | 79 | 81 | 70 | 44 | |
| 20 m sprint (s) | 3.62 ± 0.21 | 3.62 ± 0.21 | 3.57 ± 0.20 | 3.54 ± 0.23 | $p = 0.218$; partial $\eta^2 = 0.017$ |
| n | 82 | 80 | 72 | 48 | |
| $VO_2max$ (ml·kg$^{-1}$·min$^{-1}$) | 36.57 ± 5.69 | 38.06 ± 5.39 | 37.60 ± 5.45 | 38.25 ± 5.97 | $p = 0.284$; partial $\eta^2 = 0.014$ |

APHV = age at peak height velocity; CODS = change of direction speed. * Statistical significance at $p < 0.05$.



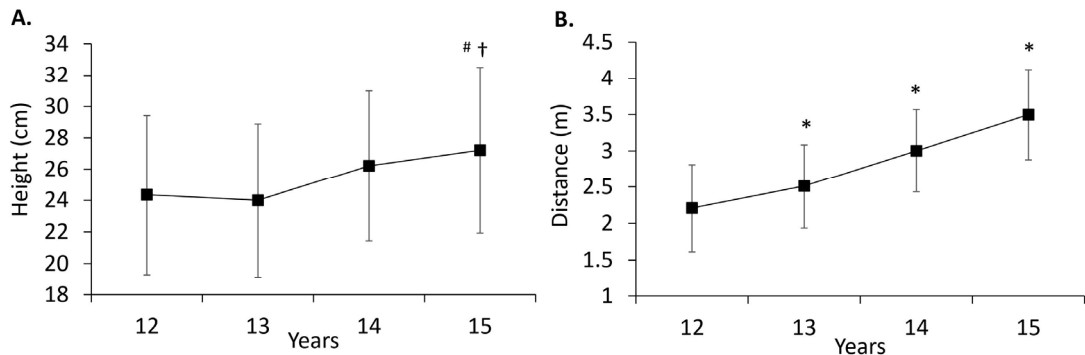

**Figure 1.** CMJ height (**A**) and MBCT distance (**B**) across age groups. Data are presented as mean ± SD. Asterisk (*) indicates significant differences to earlier and later age groups; dagger (†) indicates significant difference compared to 13 years; and number sign (#) indicates significant difference compared to 12 years. Statistical significance set at $p < 0.05$.

### 3.2. Effects of Playing Position on Participant Characteristics and Performance

A notable position effect for height emerged, as evidenced by post hoc analyses revealing increased height for circle positions in comparison to non-circle positions ($p = 0.046$, partial $\eta^2 = 0.014$) (Table 2). No significant age × position interaction was detected for height. A position effect for body mass was also evident, with post hoc analyses highlighting greater body mass for circle positions compared to non-circle positions ($p < 0.001$, partial $\eta^2 = 0.040$) (Table 2). Intriguingly, an age × position interaction emerged for body mass ($p = 0.022$, partial $\eta^2 = 0.032$), specifically at 14 years ($p = 0.017$) and 15 years ($p < 0.001$), where circle positions exhibited greater body mass.

No significant position effect or age × position interaction was observed for CODS, CMJ, and MBCT (Table 2). For sprint performances, both the 10 m sprint ($p = 0.042$, partial $\eta^2 = 0.016$) and 20 m sprint ($p = 0.010$, partial $\eta^2 = 0.025$) exhibited significant position effects, revealing faster times for non-circle positions relative to circle positions (Figure 2A and 2B, respectively). No significant age × position interaction was observed for either the 10 m sprint ($p = 0.738$, partial $\eta^2 = 0.005$) or the 20 m sprint ($p = 0.534$, partial $\eta^2 = 0.008$). Regarding VO$_2$max, a significant position effect was discovered ($p < 0.001$, partial $\eta^2 = 0.036$), showing superior values for non-circle positions over circle positions (Figure 2C). Notably, no age × position interaction ($p = 0.621$, $\eta^2 = 0.006$) was evident for VO$_2$max.

**Table 2.** Playing position characteristics and performances controlled for maturity based on age at peak height velocity.

|  | Playing Position | | Position Effect | Age × Position Interaction |
|---|---|---|---|---|
|  | **Non-Circle** | **Circle** |  |  |
| n | 163 | 140 |  |  |
| APHV (y) | 12.10 ± 0.46 | 11.69 ± 0.49 |  |  |
| Height (cm) | 167.53 ± 3.43 | 168.34 ± 3.31 | $p = 0.046$; $\eta^2 = 0.014$ * | $p = 0.966$; partial $\eta^2 = 0.001$ |
| Body mass (kg) | 65.80 ± 11.02 | 70.33 ± 10.63 | $p < 0.001$; $\eta^2 = 0.040$ * | $p = 0.022$; partial $\eta^2 = 0.032$ * |
| n | 136 | 126 |  |  |
| CODS (s) | 2.77 ± 0.22 | 2.81 ± 0.21 | $p = 0.172$; $\eta^2 = 0.007$ | $p = 0.738$; partial $\eta^2 = 0.005$ |
| n | 141 | 113 |  |  |
| CMJ (cm) | 25.84 ± 5.12 | 25.05 ± 4.99 | $p = 0.233$; $\eta^2 = 0.006$ | $p = 0.579$; partial $\eta^2 = 0.008$ |
| n | 151 | 130 |  |  |
| MBCT (m) | 2.86 ± 0.61 | 2.75 ± 0.59 | $p = 0.159$; $\eta^2 = 0.007$ | $p = 0.668$; partial $\eta^2 = 0.006$ |

APHV = age at peak height velocity; CODS = change of direction speed; MBCT = medicine ball chest throw; CMJ = countermovement vertical jump. * Statistical significance at $p < 0.05$.

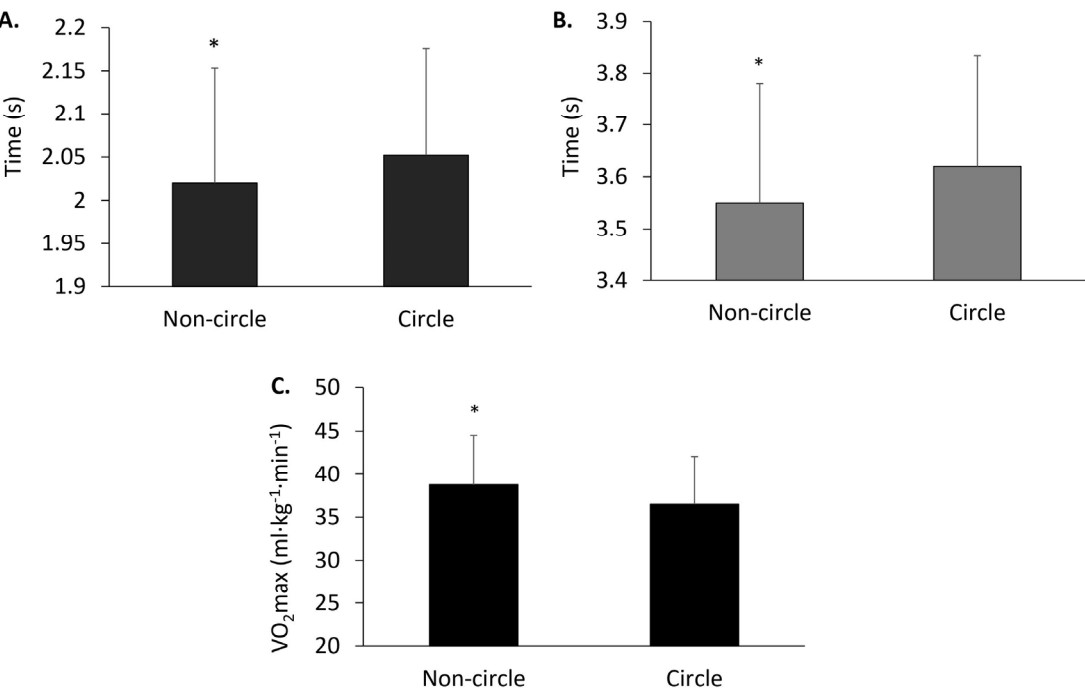

**Figure 2.** 10 m sprint (**A**), 20 m sprint (**B**), and VO$_2$max (**C**) for non-circle and circle playing positions. Data are presented as mean ± SD. Asterisk (*) indicates significant difference compared to circle position. Statistical significance set at $p < 0.05$.

## 4. Discussion

This study aimed to investigate how age and playing position impact aerobic fitness, sprint performance, CODS, and muscle power in adolescent female netball players, controlling for maturity. Age exhibited a positive correlation with MBCT performance, showing steady enhancement across consecutive age groups. CMJ performance was significantly better at 15 years compared to 12 and 13 years. However, age displayed no discernible influence on sprints, CODS, or VO$_2$max. Distinct disparities in physical fitness emerged between playing positions. Non-circle positions surpassed circle positions in sprint performance and VO$_2$max, although muscle power did not differ. Anthropometric distinctions were noted, with the circle-position players tending to be taller and heavier than their non-circle counterparts. This study underscores that, among adolescent female netball players, muscle power alone appears to be influenced by age. Furthermore, it highlights that non-circle players exhibit heightened aerobic fitness and speed relative to their circle-playing counterparts. These insights have implications for understanding physical development in young netball players and may inform training and positional considerations.

Contrary to the initial hypothesis, MBCT performance improved across successive age groups, diverging from prior research. In a cohort of 715 females aged 12–15 years engaged in diverse sports, MBCT performance improved between ages 13 and 14 years, requiring at least a two-year gap for variations in other age groups to emerge [11]. Factors like neuromuscular coordination and exercise background have been posited to influence speed and strength outcomes in adolescents [23]. Netball involves chest passes as a fundamental technique, constituting a honed skill. Specifically, players practice passing the ball quickly by propelling it from their chests using their hands and arms. Therefore, the MBCT would have been a very familiar task for the participants to perform. Since participants in our study were possibly better conditioned for MBCT, conceivably this test was more sensitive in revealing the anticipated strength changes due to muscle mass increases that tend to plateau around 15–16 years in females [24]. In the case of CMJ, though also a measure of muscle power, improvements were only conspicuous at age 15 years versus younger ages.

In netball, jumping holds significance, with up to 60 jump landings per game noted [14]. Proficiency in CMJ execution and sport-specific adaptations are pivotal for optimizing performance during this test. However, progress between consecutive age brackets might have been influenced by body composition shifts during puberty, which may not favor excelling in athletic tasks (such as increased fat mass relative to lean body mass) [25], especially when activities involve overcoming own body mass during movements like sprinting or jumping. Moreover, prior exposure to plyometric or resistance training could have influenced the differences in CMJ scores across age groups. Additionally, maturation-related increases in musculotendinous stiffness have been observed [26], playing a crucial role in enhancing muscle force production during activities involving stretch-shortening cycles, such as the CMJ. Hence, disparities in CMJ performances between age groups may be linked to variations in musculotendinous stiffness.

In contrast to the observations by Thomas et al. [27], no disparities surfaced in CMJ performance between non-circle and circle positions. This consistency resonated in our study, where MBCT results also exhibited no variation across these player roles. Notably, prior research has highlighted heightened squat jump and CMJ performance in non-circle positions compared to defenders [27]. Non-circle netball players frequently engage in more jumps and multifaceted movements during matches [28,29]. Moreover, their lighter body mass contributes to reduced inertia, potentially aiding jump execution [30]. Interestingly, our study found non-circle positions to be smaller and lighter than their circle counterparts. However, these anthropometric distinctions did not seem to impact CMJ results. A plausible rationale for the lack of observed influence on muscle power tests by playing position in the present study could be attributed to the broader age range encompassing relatively younger participants. This differs from the work of Thomas et al. [27], which solely examined a group of netballers averaging 15 years of age.

Irrespective of playing position, robust aerobic fitness holds significance for netball players [4,5,7]. Thomas et al. [12] examined the aerobic fitness of regional academy netball players using the final velocity (VIFT) achieved in the 30–15 Intermittent Fitness Test (30–15IFT). Age was a notable factor influencing VIFT, with U/17 and U/19 categories outperforming U/15. In contrast, the current study revealed that age did not impact $VO_2$max. It is conceivable that between ages 12 and 15 years, physiological changes could adversely affect aerobic fitness development in females, regardless of sporting background. A 3-year longitudinal study investigating the effect of endurance training on $VO_2$max between ages 12 and 15 years found that females experienced a slight dip in $VO_2$max relative to body mass, whereas the male scores remained unchanged [31]. Within this age period, which corresponds with puberty, there is an elevation in estrogen and other hormones in females, leading to a higher absolute fat mass accumulation compared to males (1.4 kg versus 0.7 kg per year) [32,33]. Consequently, the accentuated fat mass may impair aerobic performance, particularly running, given the need to overcome one's body mass [34].

The finding of higher $VO_2$max in non-circle positions compared to circle positions aligns with Thomas et al. [27], who reported superior aerobic fitness in non-circle positions compared to shooters and defenders (mean age of 15 years). This heightened aerobic fitness in non-circle roles likely stems from their heightened match demands, entailing greater distances covered, longer active periods, more sprints, and more frequent changes in direction compared to other positions [4,5,7,13]. The level of competition is also shown to play a role in the aerobic fitness of netballers. Elite netball players displayed superior performance in the Yo-Yo intermittent recovery test 1 (Yo-YoIRT1) compared to their sub-elite and regional counterparts [35]. This underscores the significance of cardiorespiratory fitness, not only for center-court positions but also for excelling at higher competition levels.

Netball players rely on sprinting and CODS for offensive and defensive roles [6]. Sprinting speed hinges on stride metrics and puberty-related changes like height, muscle, and lean mass [36]. Additionally, maturation of the nervous system and better muscle coordination can enhance sprint ability [37]. CODS hinges on speed, power, and body

dimensions [38], suggesting age-related improvements similar to sprinting. However, the netballers in the present study did not show age-related gains in 10 m, 20 m sprints, or CODS. Earlier research in female adolescent athletes found sprint improvements with larger age gaps (with consideration of maturation status), implying smaller gaps might not reveal sprint progress [10]. Warneke et al. [23] found no age effects on CODS (ages 10–14 years), suggesting motor control and training diversity influence results. This explanation might apply to the CODS and sprint findings in our study.

Consistent with previous research [3,27], non-circle players demonstrated superior sprint times to their circle counterparts. Surprisingly, playing positions showed no discrepancy in CODS. McKenzie et al. [3] found that non-circle compared to circle position players were faster for all sprint distances (15 m, 10 m, and 5 m) and demonstrated faster CODS assessed via the T-test. Similarly, Thomas et al. [27] identified that centers had faster sprint performances (5 m and 10 m) compared to defenders and faster 505 CODS compared to defenders and shooters. It has been suggested that faster CODS performances are strongly associated with shorter ground contact times, greater horizontal propulsive forces, and greater horizontal braking forces [39]. Furthermore, since there were no differences between playing positions for lower body muscle power (i.e., CMJ), possibly similar strength-related abilities of the lower limbs may explain why CODS did not differ between playing positions. In contrast, the differences in speed could be related to anthropometric differences, with the players in non-circle positions having lower body mass. However, future research is required to explore factors that may contribute to changes in physical fitness levels across age and/or maturity groups in adolescent female netball players.

The findings of our study should be interpreted cautiously due to acknowledged limitations. While participant maturity was considered, no information regarding training experience (e.g., sport-specific, resistance training) was collected, possibly introducing bias. Therefore, it is possible that training experience might have influenced the results of the present study. Future research should include participants' training and playing backgrounds, particularly when studying field-based testing across age groups and positions in young female netball players. Testing session scheduling is another limitation where sessions were distributed over 1–3 days due to school constraints, which might have impacted recovery time and results. Disparity in trial numbers for sprints, CODS, MBCT, and CMJ due to time constraints is a concern, potentially skewing the results. The lack of standardized language used to motivate maximal performance is an additional factor that could have influenced outcomes in tests like the CMJ. Body composition, including factors such as percentage body fat, lean body mass, and extremity (or appendicular) lean mass, was not evaluated, limiting the exploration of its potential impact on performances. The assessment of participants' maturity was solely based on somatic criteria (APHV), lacking biological parameters of pubertal development. Additionally, there was no comprehensive analysis of the stages of puberty using Tanner stages [40]. Consequently, the accuracy of the maturation assessment in our analyses may be compromised. Finally, factors like physiology, psychology, and environment could have influenced performances. In particular, nutritional status, perceived abilities, and familiarity with protocols might have confounded the results [41]. Future investigations need meticulous control over these intricate dynamics.

## 5. Conclusions

This research study showed that age-related alterations are evident in muscle power among adolescent female netball players, while no substantial changes are apparent in aerobic capacity, sprint performance, or CODS. Notably, playing positions exerted influence, revealing non-circle players have enhanced aerobic fitness and faster sprints. Circle-position players displayed greater height and body mass, potentially advantageous for their role. These findings underscore how physical fitness metrics may evolve disparately and can be shaped by the playing positions of young female netball players. Coaches should not worry if sprinting, CODS, and aerobic fitness do not progress consistently across seasons. Earlier

improvement in these fitness components is plausible with a well-designed evidence-based training program for female athletes. Considering factors like diet and psychological and musculoskeletal health is wise, as they might influence training responses. Coaches can utilize the specific age and playing position characteristics of adolescent netballers to guide player selection and tailor focused training programs.

**Author Contributions:** Conceptualization and project administration by D.A.H. and R.H.S.; supervision of project progress by R.H.S.; recruitment of subjects, data acquisition, and data processing by R.H.S.; data analysis and visualization by D.A.H.; writing and preparation of the original draft by D.A.H., K.L.W., D.L.T., and R.H.S. All authors have read and agreed to the published version of the manuscript.

**Funding:** This research received no external funding.

**Institutional Review Board Statement:** This study was approved by the University of Sydney. Human Research Ethics Committee (approval number: 2015/878).

**Informed Consent Statement:** Informed consent was obtained from all subjects involved in the study.

**Data Availability Statement:** Data will be made available on reasonable request to the corresponding author.

**Conflicts of Interest:** The authors declare no conflicts of interest.

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
