# Peer review of "Effects of Age and Playing Position on Field-Based Physical Fitness Measures in Adolescent Female Netball Players"

_pediatrrep, doi:10.3390/pediatric16010008_

Round 1

Reviewer 1 Report

Comments and Suggestions for Authors

COMMENTS ON THE MANUSCRIPT

=====================
Comments
=====================

General comment:
The authors investigated the effect of age and playing position, controlling for maturity, on physical fitness indicators in 303 adolescent female netball players aged 12.0 to 15.9 years. The manuscript is well written and the findings are of interest as they highlight the importance of controlling for maturity when assessing adolescent athletes. However, there are some sections of the paper that need editing for clarity.

Major corrections:

1.     Abstract and Keywords

Comment 1a. In the end of the abstract a recommendation for practitioners should be added.

Comment 1b. I suggest adding “maturation” in the key words as your results are controlled for maturity.

2.     Materials and methods.

Comment 2a. What was the participants’ training experience? Did they have prior experience in resistance training that might have influenced the testing outcomes? Were the participants familiarized with testing procedures prior to testing?

Comment 2b. P3 L140-141. The height of photocell gates should be mentioned based in relative bibliography.

Comment 2c. P4 L1474-175. Cues have significant impact on jump performance. What was the command used by the instructors concerning jump execution? Jump as high as you can, Jump as fast as you can or both?

Comment 2d. Report also test-retest reliability scores for CMJ and Medicine Ball Throw

3.     Results

Comment 3a. P5 L211 Please replace “weight” with “body mass” though the whole manuscript.

4.     Discussion.

Comment 4a. L285-288: Please discuss more the fact that Medicine ball throw may be considered a sport-specific test for netball athletes.

Comment 4b. P8 L291-297: Please discuss that if participants had previous experience in systematic plyometric or resistance training, CMJ scores might differ by age. Furthermore, the impact of musculotendinous stiffness - which increases through adolescence – on CMJ performance should also be discussed.

Comment 4c. L306: In Table 1, non-circle athletes are in average shorter than circle athletes, please confirm and correct

Comment 4d. L310: Do not start a sentence with”Which”

Comment 4e. L339-344: Previous studies presented their results per chronological age. The covariance analysis -with maturity as covariance – which was used in this study may be the reason of the absence of age differences in sprint and COD scores. This should be discussed accordingly.

Comment 4e. L365: to explore factors that may contribute to changes in physical fitness levels across age groups in adolescent female netball players” please write “across age and/or maturity groups”

5.     Conclusions

Comment 5a. P9 L388: A practical application of the results and a recommendation for practitioners should be added.

Author Response

Reviewer #1

We thank you for reviewing our manuscript and for the constructive comments which have enabled us to improve the manuscript.

Comment 1: Abstract and Keywords - In the end of the abstract a recommendation for practitioners should be added.

Response: Thank you for this suggestion. We have now included a recommendation for practitioners (see below).

“Coaches may be able to utilize the distinct age and playing position traits of adolescent netballers to inform player selection and design targeted training programs.”

Comment 2: Abstract and Keywords - I suggest adding “maturation” in the key words as your results are controlled for maturity.

Response: Thank you for providing this suggestion. We have now added “maturation” as a keyword.

Comment 3: Materials and methods - What was the participants’ training experience? Did they have prior experience in resistance training that might have influenced the testing outcomes? Were the participants familiarized with testing procedures prior to testing?

Response: Information about the training experience of participants was not collected and has been included as a limitation of the study in the Discussion section. Therefore, we are unable to provide any details about experience in resistance training (please see below).

“While participant maturity was considered, no information regarding training experience (e.g., sport specific, resistance training) was collected, possibly introducing bias. Therefore, it is possible that training experience might have influenced the results of the present study.”

All participants were familiarized with testing procedures prior to testing and this information has now been included in the methods (see below).

“Participants completed at least one practice attempt of the tests to become familiarized prior to commencing the actual attempts.”

Comment 4: Materials and methods - P3 L140-141. The height of photocell gates should be mentioned based in relative bibliography.

Response: Thank you for this suggestion. The following information has now been included.

“The height of the photocells was adjusted to ensure that the beam would break at the participants’ torso, in agreement with manufacturer’s instructions.”

Comment 5: Materials and methods - P4 L1474-175. Cues have significant impact on jump performance. What was the command used by the instructors concerning jump execution? Jump as high as you can, Jump as fast as you can or both?

Response: While we acknowledge that the specific wording of encouragement might influence jump performance, we are unable to provide any specific wording used for the verbal encouragement. We have listed this factor as a limitation in the Discussion section (see below).

“The lack of standardized language used to motivate maximal performance is an additional factor that could have influenced outcomes in tests like the CMJ.”

Comment 6: Materials and methods - Report also test-retest reliability scores for CMJ and Medicine Ball Throw

Response: We have now reported the test-retest reliability scores for these tests (see below).

“The CMJ has demonstrated robust test-retest reliability among adolescents, with an ICC of 0.96 [19].”

“In healthy undergraduate students the MBCT has exhibited strong test-retest reliability, with an ICC of 0.97-0.99 [20].”

Comment 7: Results - P5 L211 Please replace “weight” with “body mass” though the whole manuscript.

Response: We have made these corrections throughout the manuscript.

Comment 8: Discussion - L285-288: Please discuss more the fact that Medicine ball throw may be considered a sport-specific test for netball athletes.

Response: We agree and have added the following.

“Netball involves chest passes as a fundamental technique, constituting a honed skill. Specifically, players practice passing the ball quickly through propelling it from their chest by using their hands and arms. Therefore, the MBCT would have been a very familiar task for the participants to perform. Since participants in our study were possibly better conditioned for MBCT, conceivably this test was more sensitive with revealing the anticipated strength changes due to muscle mass increases that tend to plateau around 15-16 years in females [24].”

Comment 9: Discussion P8 L291-297: Please discuss that if participants had previous experience in systematic plyometric or resistance training, CMJ scores might differ by age. Furthermore, the impact of musculotendinous stiffness - which increases through adolescence – on CMJ performance should also be discussed.

Response: Thank you for this suggestion. We have added the following information (please see below).

“Moreover, prior exposure to plyometric or resistance training could have influenced the differences in CMJ scores across age groups. Additionally, maturation-related increases in musculotendinous stiffness have been observed, playing a crucial role in enhancing muscle force production during activities involving stretch-shortening cycles, such as the CMJ. Hence, disparities in CMJ performances between age groups may be linked to variations in musculotendinous stiffness.

Comment 10: Discussion L306: In Table 1, non-circle athletes are in average shorter than circle athletes, please confirm and correct.

Response: Correction made.

Comment 11: Discussion. L310: Do not start a sentence with”Which”

Response: Correction made.

Comment 12: Discussion L339-344: Previous studies presented their results per chronological age. The covariance analysis -with maturity as covariance – which was used in this study may be the reason of the absence of age differences in sprint and COD scores. This should be discussed accordingly.

Response: This is incorrect, Reference [10] is our previous study where we did control for maturation status during analyses. We have added text to this line so that it is clear (see below).

“Earlier research in female adolescent athletes found sprint improvements with larger age gaps (with consideration of maturation status), implying smaller gaps might not reveal sprint progress [10].”

Comment 13: Discussion L365: “to explore factors that may contribute to changes in physical fitness levels across age groups in adolescent female netball players” please write “across age and/or maturity groups”.

Response: Correction made.

Comment 14: Conclusion P9 L388: A practical application of the results and a recommendation for practitioners should be added.

Response: We agree and have now added this information (please see below).

“Coaches should not worry if sprinting, CODS, and aerobic fitness does not progress consistently across seasons. Earlier improvement for these fitness components is plausible with a well-designed evidence-based training program for female athletes. Considering factors like diet, psychological and musculoskeletal health is wise, as they might influence training responses. Coaches can utilize the specific age and playing position characteristics of adolescent netballers to guide player selection and tailor focused training programs.”

We trust that the issues above have been addressed and clarified sufficiently and we look forward to hearing from you in the near future.

Sincerely,

Dr Daniel Hackett

Reviewer 2 Report

Comments and Suggestions for Authors

Basic reporting

The authors present a cross sectional study in which they investigated the impact of age and playing position, controlling for maturity, on physical fitness indicators in 303 adolescent female netball players. Whilst the study undoubtedly has merit, there are aspects that need clarification, to improve the readability of the manuscript

ABSTRACT

Please, report the effect sizes in the abstract.

Moreover, symbols p should be in italics.

INTRODUCTION

The introduction is well-written and presents the are well and leads nicely to the main study rationale. However, what is the novelty of this study? The authors have to find some points that their study provides something new to the literature (and highlight them) and also search deeper into the literature.

MATERIAL AND METHODS

Study design

Was the order of tests randomized? Which procedure did you use to randomize the order? How the fatigue could have affected the results?

Participants

·       I suggest you improve the description of the sample selection. How have you chosen these samples?

·       It is a representative sample? If is it, introduce please the Sample size calculation. I suggest using G-Power software or similar.

·       Which was the physical activity level and/or of those participants? Do you have any data to provide?

Measurements

General comment: The main aspect is related to data acquisition. There is needed to present and detail some of the procedures before ex’plaining the tests protocol.

·       Please present information about how and where tests were performed (if some kind of warm-up was performed, who organized and applied the tests, etc.). Please detail all the main information and the necessary details in order to provide the reader with a clear picture of how physical tests were performed.

Satatistical analysis

Line 192-194: Why five groups and not four?

RESULTS

Table 1: Please, provide more information about the sample (BMI and Training experience).

CONCLUSIONS

The conclusion section should be reorganized and further developed. Moreover, a practical implications section should be further developed. How can these findings help coaches and practitioners? Which practical implications can it have in training sessions?

Author Response

We thank you for reviewing our manuscript and for the constructive comments which have enabled us to improve the manuscript.

Comment 1: Please, report the effect sizes in the abstract.

Response: We have now added effects sizes to Abstract.

Comment 2: Moreover, symbols p should be in italics.

Response: We have now italicized p values throughout manuscript.

Comment 3: Introduction - The authors have to find some points that their study provides something new to the literature (and highlight them) and also search deeper into the literature.

Response: We strongly disagree with this comment. Please review the Introduction carefully because we have emphasized that the study will expand upon previous research in adolescent netballer through investigating  “…..wider age range and greater variety of fitness measures could provide deeper insights concerning physical fitness profiles of adolescent netball players.”

Comment 4: Was the order of tests randomized? Which procedure did you use to randomize the order? How the fatigue could have affected the results?

Response: There was no randomization of testing. Please review what has been stated: To mitigate the potential impact of fatigue on test results, a strategic sequencing was employed. In instances where multiple tests were administered on a single day, the 20MSRT was consistently conducted last.”

 Fatigue was not a factor since the testing was conducted with at least 30-40 minutes between stations.

Comment 5: I suggest you improve the description of the sample selection. How have you chosen these samples?

Response: The participants were students from several high schools and involved any student involved in any sport (or even not participating in a sport). This information has already been covered “…..recruited from sports high schools in New South Wales.”

Comment 6: It is a representative sample? If is it, introduce please the Sample size calculation. I suggest using G-Power software or similar.

Response: We do not believe any power calculation is required. Please see our other publications from this study as a reference.

Hackett, D., He, W., Fleeton, J., Orr, R., Sanders, R. (2023). Effects of Age and Sex on Aerobic Fitness, Sprint Performance, and Change of Direction Speed in High School Athletes. Journal of Strength and Conditioning Research, 37(5), E325-E331.

Hackett, D., He, W., Orr, R., Sanders, R. (2021). Effects of age and sex on field-based measures of muscle strength and power of the upper and lower body in adolescents. Journal of Sports Sciences, 39(9), 955-960.

Comment 7: Which was the physical activity level and/or of those participants? Do you have any data to provide?

Response: We do not have this information/data.

Comment 8: General comment: The main aspect is related to data acquisition. There is needed to present and detail some of the procedures before explaining the tests protocol.

 Please present information about how and where tests were performed (if some kind of warm-up was performed, who organized and applied the tests, etc.). Please detail all the main information and the necessary details in order to provide the reader with a clear picture of how physical tests were performed.

Response: We have already provided this information. Again, check our consistency with our previous published paper from this study.

Hackett, D., He, W., Fleeton, J., Orr, R., Sanders, R. (2023). Effects of Age and Sex on Aerobic Fitness, Sprint Performance, and Change of Direction Speed in High School Athletes. Journal of Strength and Conditioning Research, 37(5), E325-E331.

Hackett, D., He, W., Orr, R., Sanders, R. (2021). Effects of age and sex on field-based measures of muscle strength and power of the upper and lower body in adolescents. Journal of Sports Sciences, 39(9), 955-960.

Comment 9: Line 192-194: Why five groups and not four?

Response: Because the participant numbers allowed five adequate groups to be formed.

Comment 10: Table 1: Please, provide more information about the sample (BMI and Training experience).

Response: We do not have training experience information and we do not believe BMI values would add anything to this paper and choose not to report this data.

Comment 11: The conclusion section should be reorganized and further developed. Moreover, a practical implications section should be further developed. How can these findings help coaches and practitioners? Which practical implications can it have in training sessions?

Response: The following has been included.

“Coaches should not worry if sprinting, CODS, and aerobic fitness does not progress consistently across seasons. Earlier improvement for these fitness components is plausible with a well-designed evidence-based training program for female athletes. Considering factors like diet, psychological and musculoskeletal health is wise, as they might influence training responses. Coaches can utilize the specific age and playing position characteristics of adolescent netballers to guide player selection and tailor focused training programs.”

Reviewer 3 Report

Comments and Suggestions for Authors

General Comments

This is an interesting study, evaluating the impact of age and playing position on aerobic fitness, sprint performance, change of direction speed (CODS), and muscle power in adolescent female netball players (n=303; aged from 12.0-15.9 years), controlling for maturity. The maximal oxygen uptake (VO2max) was assessed via the 20-m shuttle run test, sprint performance by 10-m and 20-m sprint time, CODS by the 505 test, and muscle power via the medicine ball chest throw (MBCT) and countermovement vertical jump (CMJ) height. Participants were grouped by age and playing position (non-circle and circle players). The somatic maturity was evaluated by age at peak height velocity.  The results indicated that age exhibited a positive correlation with MBCT performance, showing steady enhancement across consecutive age groups. CMJ performance was higher (p<0.05) at 15 years compared to 12 and 13 years. However, age displayed no significant influence on sprint performance, CODS, and VO2max. Distinct disparities in physical fitness emerged between playing positions - non-circle positions surpassed circle positions in sprint performance and VO2max, although muscle power did not differ. The Authors concluded that in adolescent female netball players, only muscle power is influenced by age, while non-circle players exhibit superior aerobic fitness and speed compared to circle players.

The manuscript is generally well written. However, the design of this paper should be little improved before publishing.

1.The Authors should describe more detailed in 2. Materials and Methods: (1)  The exclusion criteria of the participants, for example, how was controlled that the participants had: - No potential medical problems or a history of ankle, knee of back pathology that compromised their participation in the proposed performance tests. - No any lower extremity reconstructive surgery in the past 2 years. - No infections and signals of fatigue on assessment day. - No medication or drugs (pain drugs, etc.) that may influence the functional status during assessment. All these factors can significantly influence the results of this study. (2)  The somatic maturity by age at peak height velocity, APHV – how the data of APHV for different age groups (from 12 to 15 year-olds) were taken/collected (databases?). (3)  The measurement of antropometric parameters (height and body weight). I suggest to add paragraph „Anthropometry“ in Materials and Methods with detailed description of assessment of height and weight measurement. (4)  The measurement of countermovement vertical jump – how was selected the trial for further analysis (for example: the trial with best result by jumping height?).    

2. I suggest to include more limitations of this study at the end of the Discussion. The body composition (fat %, lean body mass, extremity (or appendicular) lean mass, muscle mass of the lower extremities, etc.) was not assessed (by DEXA or bioimpedance method) and also not associated with results of measured physical fitness parameters. The maturity of adolescent participants was assessed only by somatic (limited) criteria (APHV) without biological parameters of pubertal development and complex analysis of stages of puberty of the participants (by Tanner stages).

Specific Comments

2. Materials and Methods

2.2. Participants

Page 3. Please describe more detailed the exclusion criteria for the participants (see General Comments).

2.6. Countermovement Vertical Jump

Page 4. Please describe more detailed the selection criteria and parameter for further analysis (see General Comments).

Please add a paragraph “Anthropometry” (see General comments).

Please add a description of the assessment of somatic maturity by age at peak height velocity (APHV) (see General Comments).

4. Discussion

Page 9. Please describe more limitations of this study at the end of Discussion (see General Comments).

Author Response

Reviewer #2

We thank you for reviewing our manuscript and for the constructive comments which have enabled us to improve the manuscript.

Comment 1: Materials and Methods - The Authors should describe more detailed in (1) The exclusion criteria of the participants, for example, how was controlled that the participants had: - No potential medical problems or a history of ankle, knee of back pathology that compromised their participation in the proposed performance tests. - No any lower extremity reconstructive surgery in the past 2 years. - No infections and signals of fatigue on assessment day. - No medication or drugs (pain drugs, etc.) that may influence the functional status during assessment. All these factors can significantly influence the results of this study. 

Response: Thank you for this insightful suggestion. We can only provide the information below to address your comment.

“To be eligible to enrol in the study participants needed to be healthy which was deemed the absence of any musculoskeletal conditions and diseases. This was the primary responsibility of the educators from the high schools, however, researchers also conducted verbal pre-screening prior to tests to ensure that any risks to the health and safety of participants was minimized.”

We do not have any further information that can be provided.

Comment 2: Materials and Methods - The somatic maturity by age at peak height velocity, APHV – how the data of APHV for different age groups (from 12 to 15 year-olds) were taken/collected (databases?). 

Response: We have included the following information to address you comment (please see below).

“Anthropometric measurements were conducted using standardized procedures. Stature was determined through the 'stretch stature method' using a stadiometer. Participants stood against the stadiometer, aligning their heels, buttocks, and back while ensuring that the upper border of the ear opening and the lower border of the eye socket formed a horizontal line. After stretching upward and holding a full breath, the headboard of the stadiometer was adjusted until it firmly touched the vertex of the head. Stature was measured in centimeters, with at least two measurements taken for each subject. Similarly, sitting height measurements were obtained with the participant seated on a wooden box in front of the stadiometer. Body mass was recorded using clinical scales graduated to 0.1 kg, with at least two trials for each participant. Height, weight, and sitting height were entered into a excel spreadsheet that was created using the Mirwald et al. [15] equation to calculate age at peak height velocity (APHV).”

 Comment 3: Materials and Methods - The measurement of antropometric parameters (height and body weight). I suggest to add paragraph „Anthropometry“ in Materials and Methods with detailed description of assessment of height and weight measurement. 

Response: As per our response to Comment 3, we have now added this information to the manuscript.

Comment 4: Materials and Methods - The measurement of countermovement vertical jump – how was selected the trial for further analysis (for example: the trial with best result by jumping height?).  

Response: The greatest CMJ height was used for data analysis. This has now been added to the Materials and Methods.   

Comment 5: Discussion - I suggest including more limitations of this study at the end of the Discussion. The body composition (fat %, lean body mass, extremity (or appendicular) lean mass, muscle mass of the lower extremities, etc.) was not assessed (by DEXA or bioimpedance method) and also not associated with results of measured physical fitness parameters. The maturity of adolescent participants was assessed only by somatic (limited) criteria (APHV) without biological parameters of pubertal development and complex analysis of stages of puberty of the participants (by Tanner stages).

Response: We agree and have added the following information to the limitations of this study (please see below).

“Body composition, including factors such as percentage body fat, lean body mass, and extremity (or appendicular) lean mass, was not evaluated, limiting the exploration of its potential impact on performances. The assessment of participants' maturity was solely based on somatic criteria (APHV), lacking biological parameters of pubertal development. Additionally, there was no comprehensive analysis of the stages of puberty using Tanner stages [41]. Consequently, the accuracy of the maturation assessment in our analyses may be compromised.”

We trust that the issues above have been addressed and clarified sufficiently and we look forward to hearing from you in the near future.

Sincerely,

Dr Daniel Hackett

Round 2

Reviewer 1 Report

Comments and Suggestions for Authors

Authors have addressed adequately all the raised issues.